# Guaranteed Conservation of Momentum for Learning Particle-based Fluid Dynamics

**Lukas Prantl**
Technical University of Munich
`lukas.prantl@tum.de`

**Benjamin Ummenhofer**
Intel Labs

**Vladlen Koltun**
Apple

**Nils Thuerey**
Technical University of Munich

## Abstract

We present a novel method for guaranteeing linear momentum in learned physics simulations. Unlike existing methods, we enforce conservation of momentum with a hard constraint, which we realize via antisymmetrical continuous convolutional layers. We combine these strict constraints with a hierarchical network architecture, a carefully constructed resampling scheme, and a training approach for temporal coherence. In combination, the proposed method allows us to increase the physical accuracy of the learned simulator substantially. In addition, the induced physical bias leads to significantly better generalization performance and makes our method more reliable in unseen test cases. We evaluate our method on a range of different, challenging fluid scenarios. Among others, we demonstrate that our approach generalizes to new scenarios with up to one million particles. Our results show that the proposed algorithm can learn complex dynamics while outperforming existing approaches in generalization and training performance. An implementation of our approach is available at `https://github.com/tum-pbs/DMCF`.

## 1 Introduction

Learning physics simulations with machine learning techniques opens up a wide range of intriguing paths to predict the dynamics of complex physical phenomena such as fluids [28, 23, 48, 37]. As traditional simulators for fluids employ handcrafted simplifications [31] and require vast amounts of computational resources [44], recent learning-based methods have shown highly promising results. In face of the impressive performance and accuracy of these techniques, it is surprising to see that they still neglect some of the most basic physical principles, like the symmetries that arise from Newton's third law of motion and the conservation of momentum. This makes learned simulators less predictable and can lead to severe implications when basing decisions on the simulation outcomes.

Predicting physical properties with neural networks is commonly treated as a regression problem [27, 51, 54], where the training signal is defined as a soft constraint. This simple and desirable formulation allows to effectively learn and approximate physical processes but also gives way to unwanted shortcuts that deviate from the basic laws of physics. At the same time network architectures are usually designed with generalizability in mind, e.g., with applications ranging from geometry processing [34] to physical simulations [2]. In contrast, we propose a lean and efficient architecture that provides a stable and large receptive field, while adhering to the desired physical constraints. At training time, we dynamically take the unrolled evolution of the physical state into account to ensure stability when training with longer sequences.

36th Conference on Neural Information Processing Systems (NeurIPS 2022).

For the implementation of the inductive bias, our approach is inspired by *Smoothed Particle Hydrodynamics* (SPH) methods [13, 20], where physical properties are preserved by the appropriate design of the smoothing kernels [5]. Our method uses learned convolution kernels that have inherent benefits over general graph structures in the context of physics simulations, as they can easily process positional information. Instead of using a soft constraint, e.g., by integration into the loss function, our method enforces physical properties by using inductive biases while keeping the simple training signal of a regression problem. To this end, we integrate symmetries, which are an integral part of the underlying physical models, into our neural network. This results in guaranteed conservation of momentum for our method.

## 1.1 Related Work

Numerical computation of fluid-based dynamics is a long and extensively treated topic in research [15, 55]. We discuss the most relevant work here, and refer to surveys [21] for a full review. Our approach uses a Lagrangian viewpoint, where fluids are represented as smoothed particles. This viewpoint by design conserves the mass of the fluid, as the number of particles does not change, and limits valuable computation time to where the fluid is. This idea stems from astrophysics [13] and is known as *Smoothed Particle Hydrodynamics* (SPH). It was followed by a large number of papers that improved the SPH method in terms of accuracy and performance [3, 40, 26, 4, 17, 1].

In contrast to many classic simulators that use sophisticated and handcrafted model equations to describe the motion of fluids, our method belongs to the class of data-driven solvers, which entirely learns their dynamics from observations. A seminal example of learning fluid simulation from data is Ladicky et al. [22], which uses carefully designed density features with random forests to regress fluid motions. However, the method does not guarantee momentum conservation. In recent work, algorithms have been proposed to use physical priors based on Lagrangians [35, 25, 9] or Hamiltonians [36] to improve the physical accuracy of learning ODEs and PDEs, and compliance with conservation laws. Nevertheless, the methods are limited to low-dimensional problems and do not address complex PDEs such as those required for fluids. Other works have used convolutional neural networks, or ConvNets, to achieve acceleration for grid-based fluid simulations [46, 47, 19]. The former proposes ConvNets to accelerate the expensive pressure correction step [46], while the other two propose ConvNets learning corrections to reduce the grid size and, consequently, the runtime of simulations. Although these methods show large speed-ups, they inherently cannot be applied to particle-based simulations.

For processing point clouds, the community has followed multiple strategies to address the challenges with this representation. A key challenge is the permutation invariance of particles. An approach to address this uses a combination of point-wise MLPs, order-independent set functions, and farthest point sampling to create a point cloud hierarchy [34]. Another line of work processes point clouds with graph neural networks. These methods represent fluid particles with graph nodes and define edges between them [2, 23, 37]. Here, interaction networks with relation-centric and object-centric functions were developed to predict future object states [2]. Since the framework aims to be general and employs unconstrained MLPs, it cannot be guaranteed that interactions are symmetric. The same holds for Li et al. [23], which uses the same framework and builds a dynamic graph connecting nearby particles to model fluids. SPH simulations can also be treated as message-passing on graphs [37, 30, 7]. These approaches typically decompose a simulator into multiple MLPs, which act as encoder, processor, and decoder. The processor performs multiple rounds of general message-passing, updates node and edge features, and does not enforce physical properties like momentum conservation. Further, graph-based methods are strongly tied to the chosen particle discretization, and changes to it, like the sampling density, may require non-trivial changes to the models.

As an alternative to explicit graph representations, convolutional architectures represent an interesting option for particle-based simulation. Schenck et al. [39] implement special layers for particle-particle interactions and particle-boundary interactions. The work does not aim to learn a general fluid simulator but implements a differentiable position-based fluids (PBF) [26] solver, which is used to estimate liquid parameters or train networks to solve control problems by backpropagation through the solver. Moreover, Fourier neural operators [24] or their adaptive variants [14] have been proposed to solve PDEs. Most related to our approach is Ummenhofer et al. [48], which implements a trainable continuous convolution operator resulting in compact networks. The learned kernels describe general continuous functions with a smooth radial falloff but do not account for symmetric interactions of

particles. We extend the continuous convolutions to enable simulations that guarantee momentum conservation and introduce a particle hierarchy for better accuracy and robustness.

Symmetries play an essential role in physics, as isolated systems are invariant to certain transformations, i.e., the behavior of a physical system does not depend on its orientation. For example, Lie point symmetries were used to optimize data augmentation and thus improve the sample efficiency for learning PDEs [6]. Similar observations apply for the structural analysis of molecules [11] and vision applications [10, 41]. The chemical properties of proteins do not change under rotation, nor should class labels of objects in images. Rotationally equivariant architectures have been proposed to leverage this concept for improved generalization and sample efficiency, e.g., constraining the convolution kernels for the $E(2)$ group to achieve equivariance [53]. A similar approach by Wang et al. [52] was applied to physics-based problems. Kernel constraints can also be transferred to continuous convolutions on 2D point clouds for improved sample efficiency of traffic trajectory predictions [50]. For 3D problems, spherical harmonics can be used to define rotationally equivariant filters [45], which shows their application to shape classification, molecule generation, and simple problems in classical mechanics. The method guarantees rotational equivariance, but filter evaluations can become expensive for higher degrees. In addition, architectures for symmetries in arbitrary dimensions were proposed [49, 38], which however require graph-based representations.

## 2 Method

Given a physical quantity $P_t \subseteq \mathbb{R}^d$ at a time $t$ in the form of particles, let $x_t \in P_t$ denote the position, $v_t \in \mathbb{R}^d$ the corresponding velocity and $d$ the spatial dimension. The central learning objective is to approximate the underlying physical dynamics to predict the state of $P_{t+\Delta t}$ after a period of time $\Delta t$. We integrate the velocities $v_t$ and positions $x_t$ of the physical system in time while taking into account external forces $F_{ext}$, e.g., gravity, as illustrated in Fig. 1. For an initial prediction, an explicit Euler step of the form $v'_{t+\Delta t} = v_t + \Delta t \frac{F_{ext}}{m}$ and

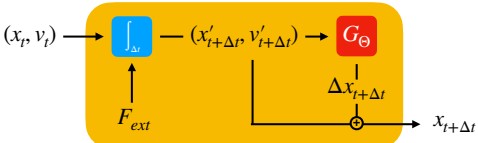

Figure 1: Each time step of our method performs a predictive time integration step (blue) for position $x_t$ and velocity $v_t$ of a set of particles at time $t$. A neural network $G$ (red) provides a position update to obtain $x_{t+\Delta t}$.

$x'_{t+\Delta t} = x_t + \Delta t\, v'_{t+\Delta t}$, is used, where $m$ represents the mass of the particles. The resulting physics state is passed to a neural network $G$ with trainable parameters $\Theta$, which infers the residual of the position $\Delta x_{t+\Delta t}$ such that particles match a set of desired ground-truth positions $y_{t+\Delta t}$. Velocities are obtained with a position-based update. This yields:

$$x_{t+\Delta t} = x'_{t+\Delta t} + \Delta x_{t+\Delta t} \;; \quad v_{t+\Delta t} = \frac{x_{t+\Delta t} - x_t}{\Delta t}. \tag{1}$$

Where the minimization target of the neural network $G$ is as follows:

$$\min_{\Theta} L(x'_{t+\Delta t} + G(x'_{t+\Delta t}, v'_{t+\Delta t}, \Theta), y_{t+\Delta t}), \tag{2}$$

with $L$ denoting a distance-based loss function, and $y_{t+\Delta t}$ the ground-truth positions. By separating the external forces and the update inferred by the neural network, the conservation of momentum only depends on the latter. In the next section, we derive which properties the network must fulfill to achieve this.

### 2.1 Conservation of Momentum

The linear momentum of a physical system is given by $M = \int_{x \in P} m_x v_x,$, where $P$ represents an arbitrary control volume in the system, $m_x$ the mass, and $v_x$ the velocity of particle $x$. It follows that the rate of change of the momentum in the absence of external forces $F_{ext}$, according to Newton's second law $m_x a_x = F_{ext} - F_x$, is given by:

$$M' = \int_{x \in P} m_x a_x = -\int_{x \in P} F_x, \tag{3}$$

where $F_x$ represents the internal forces of the physical system, and $a_x$ denotes acceleration. Hence the central condition for preserving linear momentum is

$$\int_{x \in P} F_x = 0. \tag{4}$$

It is theoretically possible to include this conservation law as a minimization goal. However, this would merely impose a soft constraint that needs to be weighed against the other terms of the learning objective. We instead propose to use the conservation of momentum as an inductive bias at training time in the form of a hard constraint.

In order to guarantee that momentum is conserved, we work with convolutional layers with specially designed kernels. We define the convolution in the continuous domain as

$$(f * g)(x) = \int_{y \in P_Q} f(y)g(y - x), \qquad x \in P_D, \tag{5}$$

where $g$ represents a learnable kernel function and $f$ is the feature vector of a quantity that should be processed in the convolution operation. Here $P_D$ denotes a set of data points, while $P_Q$ denotes a set of query points on which the convolution is performed. The use of convolutional layers already guarantees permutation invariance and translation equivariance.

However, instead of just relying on this learned kernel function, we additionally modify the convolution to further assist in ensuring conservation of momentum. For the deduction, we reformulate the residual position $\Delta x_{t+\Delta t}$ that the network generates for a given position $x_t$ as the result of an internal force $F_x$:

$$F_x = m \frac{\Delta x_{t+\Delta t}}{\Delta t^2}. \tag{6}$$

In general, the internal force at the position $x$ can be expressed as the integral of all pair-wise interaction forces exerted on $x$, i.e. $F_x = \int_{y \in P} F_{xy}$. Following Eq. 4, to enforce conservation of momentum we have to ensure

$$\int_{x \in P} \int_{y \in P} F_{xy} = 0. \tag{7}$$

If it is guaranteed that for every internal force $F_{xy}$ a corresponding opposing force exists, i.e.

$$F_{xy} = -F_{yx}, \qquad \forall x, y \in P, \tag{8}$$

then the integral of Eq. 7 evaluates to zero, and the condition for the conservation of momentum is fulfilled. This corresponds to an antisymmetry of the internal forces in the fluid.

Applied to convolutional layers, we achieve this property by adjusting the convolution Eq. 5 as follows:

$$(f * g_s)(x) = \int_{y \in P_Q} (f(x) + f(y))g_s(y - x), \qquad x \in P_D, \tag{9}$$

where $g_s$ is an antisymmetric kernel with the restriction that

$$P_D = P_Q, \tag{10}$$

and $f(x)$ is the feature vector at the position $x$, where the convolution is evaluated. The condition 10 is necessary to guarantee that there is a bidirectional connection between two points from the two sets. This is only possible if both points exist in both sets. As before, let us consider the relative features $f_{xy} = (f(x) + f(y))g_s(y - x)$, isolated for two points in space $x, y \in P$. The kernel $g_s$ is antisymmetric, i.e. $g_s(x) = -g_s(-x)$. From this follows:

$$f_{xy} = (f_x + f_y)(-g_s(x - y)) = -f_{yx}. \tag{11}$$

Due to the restriction from Eq. 10 $f_{yx}$ exists for every $f_{xy}$ and hence Eq. 8 is satisfied.

## 2.2 Antisymmetric Continuous Convolution

To implement the convolutional layer fulfilling the constraints above, we make use of continuous convolutions (CConv) [48]. While the constraints could likewise be realized with other forms of convolutional layers, CConvs are adapted for efficiently processing unstructured data, such as particle-based data from fluids, and hence provide a suitable basis to demonstrate the impact of the antisymmetric constraints. CConv layers determine the nearest neighbors from a given set of data points $P_D$ for a given set of query points $P_Q$ and aggregate their features $f$. Thereby, the features of the neighbors are weighted with a kernel function depending on their relative position. The kernel functions themselves are discretized via a regular grid with a spherical mapping and contain the learnable parameters.

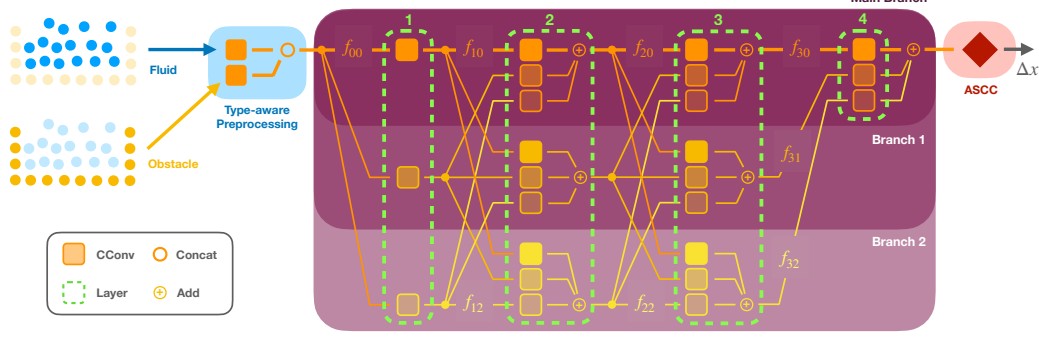

Figure 3: Neural network architecture: The colored squares symbolize the different CConv blocks, whereas the rotated square in red represents the antisymmetric layer. The color shows the used query point set. Orange corresponds to the original point set, with each set below halving the resolution. Search radii are enlarged accordingly.

The original CConvs are defined as: $\text{CConv}_g(f, P_D, P_Q) = (f * g)(x) = \sum_k f_k g(x_k - x)$ for $k \in \mathcal{N}_r(P_D, x)$ and $x \in P_Q$. This corresponds to a discretized version of Eq. 5, where $\mathcal{N}_r$ represents the nearest neighbor search for a fixed radius $r$. To conserve momentum, we adapt the equation according to Eq. 9 as follows:

$$\text{ASCC}_{g_s}(f, P_D, P_Q) = (f * g_s)(x) = \sum_{k \in \mathcal{N}_r(P_D, x)} (f + f_k) g_s(x_k - x), \qquad x \in P_Q, \qquad (12)$$

with the restriction that $P_D = P_Q$. Here the discrete, antisymmetric kernel $g_s$ consists of a grid with a user-defined size. To obtain the antisymmetric property we halve the learnable kernel parameters along a chosen axis and determine the second half by reflection through the center of the kernel. Additionally, the mirrored values are negated, as shown in Fig. 2. Mathematically, the mirroring fulfills Eq. 9, and hence the axis of reflection for the kernel weights can be chosen freely. The convolution is evaluated with the reflexive term $f + f_k$ from Eq. 9 to preserve the antisymmetry. In the following, we refer to the proposed *antisymmetric CConv* layer as *ASCC*.

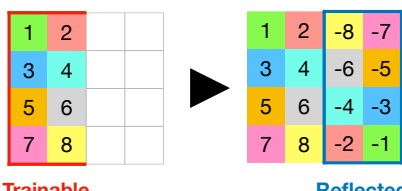

Figure 2: The trainable variables are negated and mirrored by the center point. This results in an antisymmetric kernel.

## 2.3 Neural Network Formulation

To construct a full neural network model that preserves momentum, we build on a CConv architecture, replacing the output layer with an ASCC layer, as shown in Fig. 3. We deliberately replace only the last layer, as this maximizes the flexibility of the previous layers and still guarantees momentum preservation. While antisymmetric kernels have highly attractive physical properties, they intentionally restrict the scope of actions of a neural network. The resulting networks generalize much better, but the initial learning objective is more complex than for unconstrained networks. Regular networks can take unphysical shortcuts by overfitting to problem-specific values. A corresponding evaluation of the generalization capabilities of ASCCs is provided in Sec. 3.

**Layer Hierarchy** To support learning generalizable physical dynamics, we employ a hierarchical network structure that allows for a large receptive field. Specifically, we use an architecture inspired by the parallel processing of the multi-scale feature aggregator of HRNet [43]. It processes sets of extracted features at different spatial scales with separate branches of the network. Akin to pooling layers, the density of the sampling points is reduced successively in each branch. The structure of the network and its branches are shown in Fig. 3.

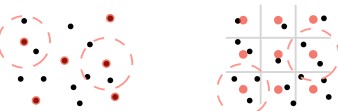

Figure 4: Farthest point sampling (left) vs. voxel sampling (right). Data points in black, sampling points in red. The red circles indicate sampling region's for feature aggregation.

The main branch of the network works on the full set of sample points $P_{Q0}$ which is in this case equal to the data points $P_D$. For the secondary branches, a new set of sampling points $P_{Qi}$ is

generated from the input points $P_D$ with a scaling factor $s_i$ for the corresponding branch $i$ such that $s_i \approx \rho(P_{Qi})/\rho(P_{Q0})$, with $\rho$ denoting the density function of the given point set. Additionally, points in each set should be spatially distributed as evenly as possible so that the scaling factor remains similar across all partial volumes. To satisfy condition 10, the point features are resampled back to original sample points $P_{Q0}$ before processing them in the ASCC layer.

Generating the sampling points $P_{Qi}$ of a layer is a non-trivial task, for which previous work typically employs farthest-point sampling (FPS) [34]. While FPS extracts a uniformly distributed subset from a point cloud, its time complexity of $\mathcal{O}(NlogN)$ is not favorable. We have found that a *voxelization* approach is preferable, which in a similar form is already used in established computer vision methods [56, 42]. We choose the centers of the cells of a regular grid, i.e. voxels, as sampling points, as illustrated in Fig. 4. The spacing of the grid is determined from the initial particle sampling modified by the scaling factor $s_i$. The voxelization can be performed in $\mathcal{O}(N)$, and results in an evenly distributed set of sampling points.

**Feature Propagation**  To compute the features $f_{ij}$ for level $i$ and branch $j$, we use a set of CConvs that process the features of all branches from the previous pass for each branch. This results in a multi-level feature aggregation across all branches. For merging the features, the different features are accumulated via summation:

$$f_{ij} = \sum_k \text{CConv}(f_{(i-1),k}, P_{Qk}, P_{Qj}), \tag{13}$$

where $k$ denotes all branch indices of the previous layer, as also illustrated in Fig. 3. Here, it is important to choose the sampling radius of the convolution operator based on the scale of the corresponding input branch $s_k$. This ensures that the convolutions on average always process a neighborhood of similar size. I.e., for a small input resolution a larger radius is used, and vice versa. Finally, the accumulated features of the main branch are processed by an ASCC layer.

## 2.4  Training and Long-term Stability

Following previous work [23, 48, 37], we use a mean absolute error of the position values between prediction and ground truth weighted by the neighbor count as our loss function:

$$L(t) = \frac{1}{|P|} \sum_i e^{-\frac{c_i}{c_{\text{avg}}}} |x_{t,i} - y_{t,i}| \tag{14}$$

Here, $c_i$ denotes the number of neighbors for particle $i$, and $c_{\text{avg}}$ the average neighbor count. We purposefully formulate the loss to work without additional loss terms, such as physical soft constraints, and instead rely on the conservation of momentum from the ASCC.

For temporal stability, we use a rollout of $T$ frames at training time. That is, for each training iteration, we run the network for $T$ time steps, where the input for the next step is based on the result from the previous prediction. The loss is evaluated and averaged across all $T$ frames with $L_r = \frac{1}{T} \sum_{t=0}^{T} L(t)$. This integrates the temporal behavior in the training and stabilizes it for a medium time range.

While larger $T$ are preferable for improving long-term predictions, they lead to significantly increased memory requirements for backpropagation. To counteract this effect, we prepend the rollout by a sequence of $W$ simulated preprocessing steps using the current training state. Altogether $N = W + T$ frames are generated, but irrespective of the number of preprocessing steps, the loss and hence gradient, is only computed for $T$ steps after preprocessing. This lets the network learn to correct its own errors that occur over longer time spans and improves long-term stability even in complex scenarios without increasing the memory. During training, we adapt $W$ based on the training progress and the example difficulty. We provide the details of the implementation in App. A.2.

## 3  Results

We perform a series of tests to show the physical accuracy, robustness, and generalizability of our method compared to other baseline methods. We primarily use data from a high-fidelity SPH solver with adaptive time stepping [1]. The resulting, two-dimensional dataset `WBC-SPH` consists of randomly generated obstacle geometries and fluid regions. Gravity direction and strength are additionally varied across simulations. In addition to this primary dataset, we also use the MPM-based fluid dataset `WaterRamps` from Sanchez-Gonzalez et al. [37], and the three-dimensional

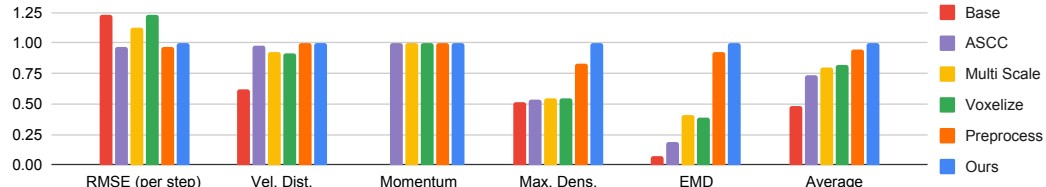

Figure 6: Quantitative evaluation for the ablation study: We consider a relative accuracy to the final method (Ours as 1.0) with higher values meaning better performance. The metrics are computed as described in Sec. 3, with *Vel. Dist.* denoting the JSD of the velocity distributions. While *RMSE (per step)* shows the RMSE after one time step, all other metrics are evaluated over a whole sequence. *Average* corresponds to the mean of all previous metrics.

liquid dataset "Liquid3d" from Ummenhofer et al. [48] for additional evaluations. Both consist of randomized fluid regions with constant gravity. A more detailed description of the datasets is provided in App. A.3.

To quantify the accuracy of the methods, we use several different metrics with respect to the ground truth data from the corresponding dataset. We use the root-mean-squared error (RMSE) for single simulation steps to evaluate short-term accuracy. However, over long sequences, it is common for fluid particles to mix chaotically regardless of whether the general fluid behavior resembles the reference, leading to artificial per-particle errors. Therefore, the earth mover's distance (EMD) is used as an assessment of the long-time accuracy over the fluid volume as a whole [12]: $\min_{\phi:x \to y} \sum_x \|\phi(x) - y\|_2^2$, where $x$ and $y$ are positions from two point sets and $\phi$ denotes a bijective mapping of minimal distances. EMD matches the particles as probability distributions, such that the global distance error is minimized. The deterministic assignment of the particles makes EMD agnostic to the ordering of particles, which counteracts the assessment errors caused by mixing. Intuitively, EMD evaluates the similarity of density-weighted volumes instead of the individual particles. Thus, it quantifies the error of the simulated configuration relative to the reference, neglecting small-scale changes in particle order. In addition, we evaluate the difference between the velocity distributions to evaluate the accuracy of the velocity content of a simulation. We compute a histogram of the magnitude of the velocities and compute its Jensen-Shannon divergence to the histogram from ground truth data: $\text{JSD}(V_x \| V_y) = \frac{1}{2}(D_{KL}(V_x \| V) + D_{KL}(V_y \| V))$, where $V = \frac{1}{2}(V_x + V_y)$ and $D_{KL}$ is the Kullback-Leibler divergence. We also evaluate deviations in terms of maximum density, $|1 - (\max_i \rho(x_i)/\max_i \rho(y_i))|$, with $\rho$ being a function to compute the density of particle $i$. This evaluation correlates with the compressibility of the fluid and gives an indication of the stability of a method. Lastly, we also compute the linear momentum change as $\sum m_i a_i - F_{ext}$, where $a_i$ is the acceleration and $m_i$ the mass of a particle, and $F_{ext}$ denotes external forces.

The source code of our approach, together with the associated evaluation tools and links to datasets, is available at https://github.com/tum-pbs/DMCF.

**Standing Liquid** We first consider one-dimensional simulations of a hydrostatic column of liquid. The training data consists of a different number of fluid particles which, under the effect of gravity, rest on a fixed boundary at the bottom. The reference stems from a solver with implicit time stepping [16] that can handle global effects instantaneously. To test the generalization, we also evaluate the methods for a free fall scene. Here we generate a certain number of fluid particles in free fall, which substantially deviates from the behavior of the training data. We trained two networks with the architecture from Sec. 2.3, once with the proposed ASCC layer (ASCC) and once with a regular CConv layer instead (No Sym). Despite the seeming simplicity of the test case, even small inaccuracies can lead to unstable behavior over time.

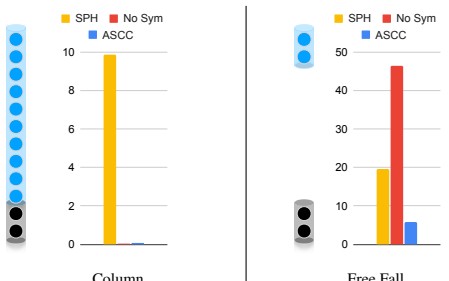

Figure 5: RMSE ($\times 10^{-3}$) evaluation over 100 steps for a set of *Column* scenes and *Free Fall* scenes. A schematic is shown on the left side of the graphs: fluid particles (blue) are affected by gravity; solid boundary particles are shown in black. The ASCC network shows the best performance in terms of generalizing to the free fall cases.

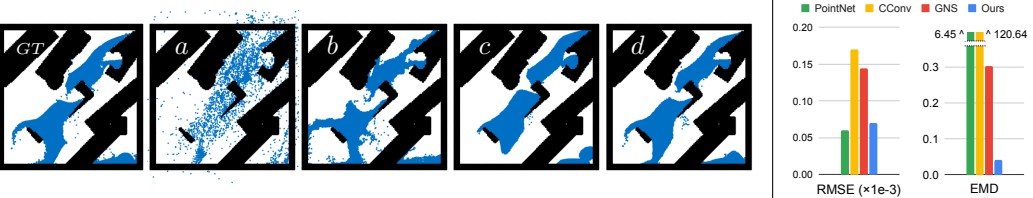

Figure 7: Excerpts from a test sequence with the high-fidelity `WBC-SPH` dataset and a quantitative evaluation on the right. The frames on the left represent, f.l.t.r., ($GT$) the ground-truth, ($a$) PointNet, ($b$) CConv, ($c$) GNS, and ($d$) Ours.

Without the antisymmetric constraint, a network can simply overfit to the gravity and negate it. This reduces the problem to a local per particle problem. With ASCC, on the other hand, the network relies on particle interactions to solve the problem. This makes it necessary to determine a global context, and infer a pressure-like counter force to stabilize the fluid. We compare the performance of both learned versions with the solution obtained by an explicit SPH solver with a small time step [32]. For new column heights, Fig. 5 shows that both learned versions quickly compensate for compression effects in the liquid and yield very small errors, while the SPH solution oscillates in place. Whereas SPH handles the free fall case on the right reasonably well with an error of $19.68$, the network without antisymmetry largely fails and produces erroneous motions with an error of $46.36$. The antisymmetric version, having learned a physically consistent response, yields an error that improves over SPH by a factor of 3.3.

**Ablation Study**    We evaluate the relevance of individual features introduced for our method with the ablation study using the `WBC-SPH` dataset. We start with a base model (*Base*) without an ASCC layer and gradually add relevant features. While this model fares very well for single steps, as shown in Fig. 6 it performs poorly with respect to all other evaluations. Here, the graphs show the performance relative to the full method, i.e., $L_{\text{Ours}}/L$, with $L$ denoting the loss of a variant under consideration. For the second version, we add the antisymmetrical layer (*ASCC*), which ensures the conservation of momentum and significantly improves the motion in terms of velocity distribution. Here the momentum score of $0.0$ for *Base* corresponds to a effective error of $0.095$, all following variants having a score of $1.0$ with zero error. Next, we evaluate the relevance of the multi-scale processing. We tested two different variants, one with FPS (*Multi Scale*) and one with the voxelization (*Voxelize*). The performance is very similar for both, with the latter variant having clear advantages in terms of resource usage. While improvements following the multi-scale handling are not obvious first it starts to more clearly pay off when combined with preprocessing for temporal coherence training in (*Preprocess*). The improved long-term stability and accuracy are clearly visible in the maximum density and EMD metrics. Lastly, our full method (*Ours*) includes additional input rotation normalization and a more advanced boundary processing step, as explained in more detail in App. A.1. This leads to further slight improvements and yields the final algorithm, which we will compare to previous work in the following sections.

**Comparisons with Previous Work**    In the following, we compare our method with several established methods as baselines: an adapted variant of PointNet [33], the CConv network [48] and GNS [37]. More information about the employed networks and the training procedures are given in App. A.2. We train and evaluate networks with our `WBC-SPH` dataset. For fairness, we additionally evaluate the different approaches with their respective data sets, i.e., with the `WaterRamps` dataset of GNS and with the `Liquid3d` dataset of CConv.

When trained on the `WBC-SPH` dataset, which consists of sequences with 3200 time steps, neither CConv nor PointNet are able to generate stable results. While the short-term RMSE accuracy is good, both networks fail to stabilize the challenging dynamics of our test sequences. Examples are shown in Fig. 7. The outputs from GNS are stable, but the network seems to maintain stability by smoothing and damping the dynamics. Also, despite receiving multiple frames over time, the GNS model has difficulty predicting the dynamics of changing gravity directions. For this scenario, the antisymmetric ASCC network clearly outperforms GNS and the other variants in quantitative as well as qualitative terms. In particular, its EMD error is 7.22 times smaller than the closest competitor.

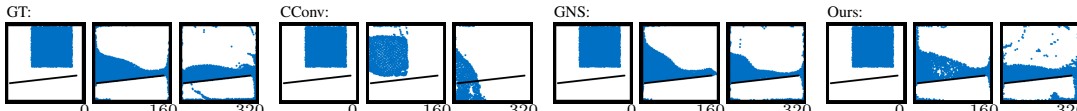

Figure 8: Snapshots over time from models for the `WaterRamps` dataset.

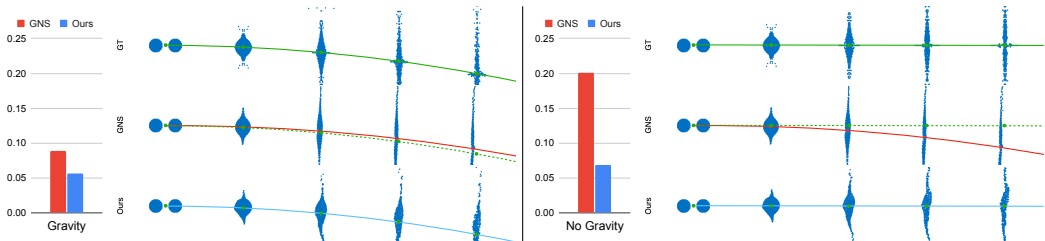

Figure 10: Assessment of generalization with droplet collisions: left with, right without gravity. Each side shows an EMD evaluation and three centers of mass trajectories with overlaid snapshots over time. The reference trajectories are shown in green. While our approach closely matches these trajectories, hiding them behind the blue lines, GNS noticeably deviates, yielding larger errors in terms of EMD.

For fairness of the comparison, we also retrain our model and the CConv model with the `WaterRamps` dataset. While the accuracy of GNS with this data is very high, our method nonetheless outperforms GNS with a decrease of 24% in terms of EMD. In comparison, CConv results in a 3.61 times larger EMD. Over short periods the dynamics of GNS behave similarly to ours. For longer periods of more than 30 frames, the results diverge, as shown in Fig. 8. Most importantly, our network yields a high accuracy while requiring only a fraction of the computational resources that GNS requires (Fig. 9): the ASCC network has 29% of the weights of GNS, and inference is 2.79 times faster, while yielding the aforementioned improvements in terms of accuracy. In addition to the conservation of momentum, this

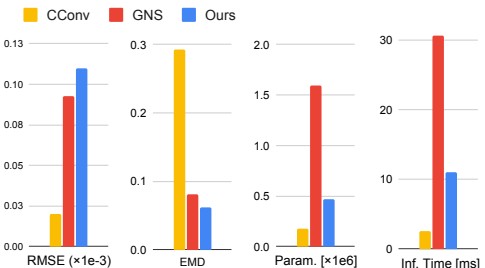

Figure 9: Quantitative evaluations with the `WaterRamps` dataset. We show the accuracy of the models along with their size and computational performance.

advantage can be explained by the inductive biases of our convolutional architecture: our networks can efficiently process positional information, which is difficult for generic graph networks. We discuss the inherent advantages of the convolution architectures over graph networks in more detail in appendix A.2.2. While CConv performs best in terms of resources, it does so at the expense of inference accuracy: its EMD is three times larger than the one obtained with our method.

**Generalization**   To assess the generalizing capabilities of our network and GNS trained with the `WaterRamps` data, we perform tests with droplets colliding under the influence of varying gravity. This case allows for an accurate evaluation in terms of the center of mass trajectory of the liquid volume. Considering a case with default gravity, the ASCC network already yields an EMD error reduced by 57%. When reducing the gravity to zero, the GNS trajectory noticeably deviates from the ground truth trajectory and yields a 195% larger error than our method. The GNS has encoded the influence of gravity in its learned representation and hence has difficulties transferring the dynamics to the changed physics environment.

**3D Simulations**   Finally, we test our network with 3D data using the `Liquid3d` dataset. Unlike the other datasets, its simulations use a significantly larger time step (8 times larger than `WBC-SPH`). Combined with the higher dimensionality, it is substantially more difficult to obtain stable solutions. Again, our method achieves comparable accuracy with much better generalization. Similar to the previous examples, it becomes apparent that CConv tends to overly dampen the acceleration of gravity, which, unlike our method, leads to discrepancies in terms of the motion compared to the ground truth, as shown in Fig. 11. To show the scalability and generalizability of our method, we apply it to multiple large-scale scenes with up to 165× more particles than in the training set. Qualitative results

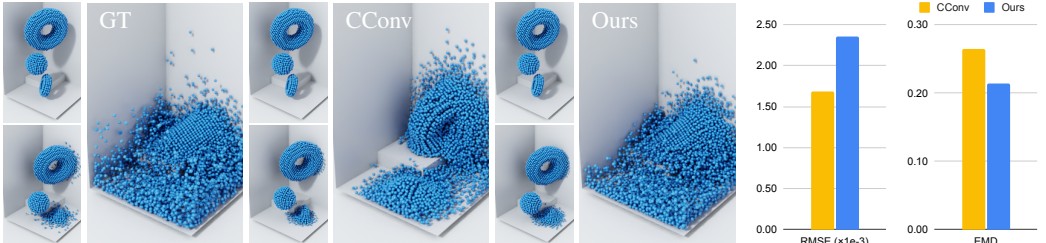

Figure 11: Example of a 3D sequence based on the `Liquid3d` dataset. CConv visibly dampens the acceleration by gravity while our method closely matches the dynamics of the ground truth.

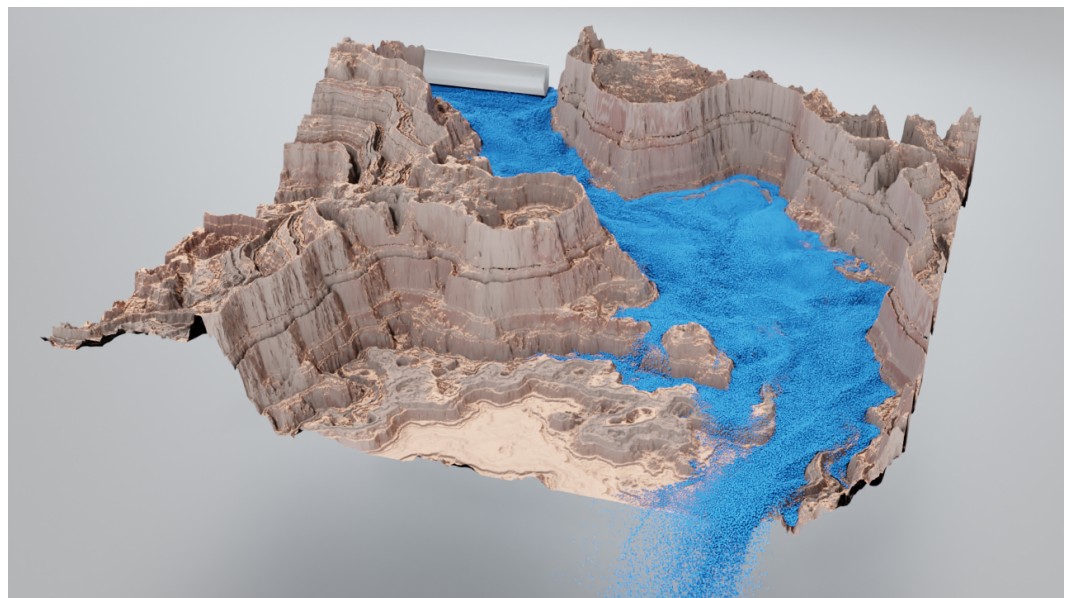

Figure 12: Test scene with over one million particles after 800 time steps generated with our method.

can be found in Fig. 12 and App. A.4 (Fig. 24). Our model robustly predicts the dynamics of these scenes using the same time step as for the smaller scenes.

**Performance** We have also evaluated the computational performance of our method, with detailed measurements given in App. A.4. E.g., for the 2D data `WBC-SPH` our method requires $67.25ms$ per frame on average. For comparison, the corresponding reference solver requires $10925ms$ per frame. Thus, our network achieves a speed-up of 162 over the reference solver.

## 4 Conclusion

We have presented a method to guarantee the conservation of momentum in neural networks and demonstrated the importance of this concept for learning the complex dynamics of liquids. In particular, our approach outperforms state-of-the-art methods in terms of accuracy and generalization. Nonetheless, the limits of symmetric constraints are far from exhausted. E.g., while our network is antisymmetric, permutation invariant, and translation invariant, an interesting avenue for future work will be to investigate more generic forms of $E(n)$ equivariance [8, 50, 38].

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
