# A Appendix

In the following sections, we provide additional details about the network architecture, training, and experiments. The source code and WBC-SPH data set are published at `https://github.com/tum-pbs/DMCF`.

## A.1 Implementation Details

We implement our neural network with Tensorflow (`https://www.tensorflow.org`), and use the Open3D library (`http://www.open3d.org`) for the continuous convolutions (CConvs). They also serve as the basis for the implementation of our antisymmetric CConv (ASCC) layer.

**Axis for Mirroring**  As mentioned in the main text, the mirror axis for ASCC layers can be chosen freely while fulfilling the requirements from theory. This provides a degree of freedom for implementation. We decided to use a fixed axis, which in our case corresponds to the spatial y-axis. While the mirroring could potentially be coupled to the spatial content of features, we found that a single, fixed axis for mirroring simplifies the implementation of the ASCCs, and hence is preferable in practice.

**Additional Modifications**  In addition to the properties of our algorithm as discussed in Section 2.3 and the ablation study in Section 3, we normalize the input data depending on the given gravitational direction in the model. We have found that this slightly improves different directions of vector input quantities. The output is denormalized again before continuing with the time integration steps. Moreover, to satisfy the condition of the constraint 10 from above, i.e. $P_D = P_Q$, it is important for the antisymmetric model to process the boundary particles in addition to the fluid particles as the input to the ASCC layer, even if only the output for the latter is used. This allows the ASCC layer to incorporate reactions to boundary conditions in its output directly, and it ensures the pairwise symmetry for 10. In this aspect, our method differs from previous methods such as CConvs, which process the boundary particles only in the input part and ignore them in the rest of the network. While it would be sufficient to add the boundary particles before the ASCC layer at the end of the model, in practice, we include the boundary particles in all layers. This affects performance due to the somewhat larger number of particles to be processed. However, in combination, the normalization and boundary handling lead to slight improvements in accuracy. We measure the influence of these two additions over the *Preprocess* model in our ablation study, which shows an increase from 0.94 to the final 1.0 (*Ours*) in terms of averaged relative performance.

**Ablation Study**  The main text mentions that the ablation study score is evaluated relative to the final version $L_{Ours}/L$. In cases where the values become zero, however, e.g., for the change of momentum, the evaluation of this term is no longer well defined. In practice, we add a small constant epsilon via: $(L_{Ours} + \epsilon)/(L + \epsilon)$. We chose $10^{-100}$ as the value for $\epsilon$ to compute the relative scores provided in the main paper.

## A.2 Training Details

For training, we use Adam as the optimizer and train with a batch size of 2 and an initial learning rate of 0.001. We use a scheduled learning rate decay and halve the learning rate in intervals of $5k$ iterations, starting at iteration $20k$. The training has a total duration of $50k$ in iterations. For all convolutions, we use the random uniform initializer with range $[-0.05, 0.05]$. For additional temporal coverage of the training, we train our model with a rollout of $N = 3$ frames. This value is increased to $N = 5$ from step $15k$ onwards. Similar to GNS [37], a noise with standard distribution is added to the training input. We use a standard deviation of size $0.1r$, where $r$ corresponds to the particle radius of the data.

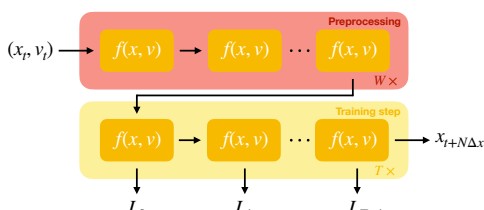

Figure 13: Rollout at training time: Each $f(x, v)$ represents one time step. After a random number of $W_{max}$ precalculation steps (red), the network is executed for $T$ time steps to compute the training loss (yellow). The gradient is evaluated only for the last part (yellow).

We ran our training on an NVidia 2080ti GPU (12GB) for the 2D data sets and with an NVidia A6000 GPU (48GB) for the 3D data set.

**Preprocessing**  As discussed in Section 2.4, we use custom preprocessing steps to improve long-time stability. We evaluate the network for a random number $W \in [0, W_{max})$ of time steps before providing the final state to the training step, as illustrated in Figure 13. $W_{max}$ is continuously increased throughout the training, as long rollouts are not meaningful in earlier stages of the training process. We enable preprocessing starting from step $10k$ with a starting value of $W_{max} = 5$. At step $20k$ and $30k$ we double the value of $W_{max}$.

Despite this scheduled increase, we found that the preprocessed states, due to their randomized nature, can lead to overly difficult states throughout the training. In the context of fluids, the maximum density of the fluid is a good indicator for problematic states. Hence, we stop the preprocessing iterations for a sample if it exceeds a chosen threshold in terms of

$$E = |1 - (\max_i \rho(x_i) / \max_i \rho(y_i))|, \tag{15}$$

with $\rho$ being a function to compute the density of particle $i$. This ensures that the states at preprocessing time do not deviate too much from the ground truth. At the same time, the threshold preserves challenging situations during which it is essential to train the network such that it learns to stabilize the state of the system.

### A.2.1  Neural Network Architecture

The neural network we employ has three distinct parts, as illustrated in Fig. 3 of our main paper. The first part *Type-aware Preprocessing* consists of several parallel CConv layers, one for each particle type in the input. We use two layers to process the two types of particles (fluid and obstacle). The feature dimension of the CConvs is 8 per particle type. As shown in previous work [48], this approach, performs better than type-specific input labels for particles. The type-specific features generated in this way are concatenated for further processing in the following layers. Additionally, type-aware handling benefits using different features for the different particle types. In addition to the spatial position, the input features are velocity and acceleration for the fluid particles and surface normals for the obstacle particles.

In the main body of our architecture, the *Multi-scale Feature Aggregation* part, the preprocessed features are passed through several layers. The number of layers in our final network is 4, each consisting of multiple CConvs working on 4 different branches with different resolutions. The first branch of these four retains the original scaling and is referred to as the main branch. In the first layer $L1$ of the feature aggregation part, we split the features into the 4 branches with 4 different CConvs. For each CConv we use different query points, which we generate using the voxelization approach. The density of the selected query points determines the resolution of the output. After this first layer, we obtain different features with different resolutions for each branch. The scaling factor of the different branches is $1, \frac{1}{2}, \frac{1}{4}$ and $\frac{1}{8}$, respectively, with corresponding radii of $r, 2r, 4r$ and $8r$, where $r$ is the particle radius of the input data. The voxel size for voxelization is given by $\frac{r}{2}$ and is also divided by each branch's corresponding scaling factor. The feature dimensions of the CConvs used in the first layer are $16, 8, 4$ and $4$, starting with the convolution of the main branch. In the second $L2$ and third $L3$ layer, the division into 4 branches is maintained. We use 4 CConvs per branch, each of which processes a multi-scale feature from the previous layer. The result is merged with an addition and corresponds to the new feature for the corresponding branch. This results in a total of 16 CConvs per layer. The feature dimension remains the same for all 4 CConvs per branch but varies with the respective branch. For both layers, we use $32, 16, 8$, and $4$ feature channels. In the fourth layer, $L4$, the branches are merged back into the main branch. The 4 CConvs have a feature dimension of 32 each.

In the final and third sections, we use the anti-symmetric ASCC as the output layer, enforcing the desired conservation of momentum. The feature dimension of the ASCC layer is determined by the desired spatial output dimension.

For all CConv layers we use a kernel size of 8 in all dimensions with a *poly6* kernel [29] as window function. The same applies to the ASCC layer used, with the difference that we use a *peak* kernel [21]. We found this prevents clustering of the particles compared to a *poly6* kernel.

The parameterization explained above is the default for the `WBC-SPH` data set. The `WaterRamps` data set instead contains smaller particle neighborhoods and less complex dynamics. Hence, for training with this data set, we use the same hyperparameters, while the number of branches is reduced to 3 by removing the branch with the lowest resolution. As with `WBC-SPH`, we use a maximal rollout of 5 in training for the model used for the generalization (Section 3) and the robustness test (Section A.4). However, for our final version for the comparison with GNS we found that a longer rollout further improves results. Here we use a rollout of 20 steps, which results in a EMD error of $0.06156$, compared to $0.09155$ for the 5-step model. This is also shown in Figure 14.

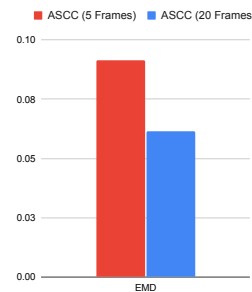

Figure 14: Accuracy comparison for our `WaterRamps` model trained with 5 and 20 steps.

The same applies to the network for the `Liquid3d` data set. Here we have additionally removed the third layer and reduced the size of the CConv kernel to $4$ and of the ASCC kernel to $6$.

**PointNet** PointNet can be seen as a fundamental baseline, similar to a fully-connected network in other problem settings. The original method was initially designed for classification instead of regression tasks. The input to PointNet is usually the complete point cloud, from which a single scalar/vector is generated. To adapt it for our problem setup, we evaluate the PointNet for all input particles, thus generating output for each particle. Additionally, we use only the considered particle's neighborhood as input.

For the PointNet [33] baseline, we used a fully-connected neural network with 5 layers that process particles individually. In order to establish a relation amongst the neighboring particles, we additionally accumulated the features of the neighboring particles after each fully-connected layer with a *poly6* kernel for each particle. The number of neurons per layer was $64, 128, 128, 128$, and $3$.

**GNS** For our tests with the GNS [37] model, we use the official implementation provided by the authors. We trained the GNS model at first with the provided `WaterRamps` data set, using the code from the original paper without modifications, for 5M iterations until the validation loss converged.

In addition, we trained a GNS model with our `WBC-SPH` data set. For this, we modified the hyperparameters of the network to fit our data set by halving the search radius to $0.0075$, setting the batch size to 1, and reducing the input noise to $3.3e^{-4}$. Again, we trained the network until the loss converged after 1.25M steps. For training, we proceed as for the `WaterRamps` data set: the GNS receives a sequence of 6 temporal frames as input from which velocity and acceleration are constructed. It is worth noting that our ASCC model only receives a single frame as input, with the current acceleration and velocity as additional features. Thus, the varying external forces of the data set can be constructed for the GNS from the data sequence, whereas for our method, the acceleration of the particles depends on it.

The set of six time steps used as input provides the GNS network with additional temporal information. According to the authors, this plays a vital role in generating stable results [37]. E.g., the network can reconstruct the acceleration acting on the particles from the provided sequence. This approach has the limitation that the six frames must be provided, pre-computed, and processed each time. In our case, we restrict the network to work with a single frame as input while the velocity and acceleration of the particles are provided explicitly as input features. E.g., this allows our method to work with a static initialization frame without the need to generate a sequence beforehand.

Another difference in terms of architecture is that the GNS does not add gravity accelerations to the model outputs, which, according to the authors, does not influence the performance. Hence, GNS learns the acceleration due to gravity for free-falling particles as well, unlike our method, where the gravitational acceleration is applied independently of the network. This has the advantage that GNS does not have to compensate for the gravitational effect in hydrostatic conditions, which we found essential for stable simulations. On the other hand, our model directly generalizes to different external forces than gravity. If necessary, these forces likewise would have to be provided as inputs.

**CConv** We likewise used the author's implementation for the CConv [48] model. When training the CConv model with our `WBC-SPH` or with the `WaterRamps` data set, we halved the search radius of the CConvs compared to the original. Similar to our method, the CConv model does not receive a

sequence of data as input, and hence we pass the acceleration as an additional feature as input to the network.

### A.2.2 Discussion: Comparing CConv and GNS

Our method builds on CConvs as central building blocks for our neural network architecture. As an alternative to CConvs, Graph Neural Networks (GNNs) provide an established framework for processing unstructured data. Even if the data is not given in the form of an explicit graph structure, a graph can be created dynamically by creating new edges based on the proximity of particles and a distance threshold. While graphs and spatial convolutions can be seen as equivalent representations that can be transformed into one another for a given discretization with particles, CConvs contain an explicit inductive bias in the form of positional information of the convolution kernel. The relative positions of query points are directly put into correlation. This is a crucial operation of classical Lagrangian discretizations such as SPH and is supported by CConvs without having to be learned and encoded in parameters. Consequently, CConvs yield leaner networks with correspondingly faster evaluation and training times. Additionally, in contrast to graphs, the kernels are regularized by construction through the discretization. This improves the generalization to different sampling densities, as we show below.

### A.3 Simulation Data Sets

The data sets for the evaluation of our method are based on particle-based fluid simulations. The data was generated with different solvers, the properties of which we discuss in detail below. As spatial units, we use meters.

**Liquid Column** For the liquid column data set, we use an iterative solver following He et al. [16] with an error threshold of $0.01$. We use a particle radius of $0.005m$ and a time step of $2.5ms$. The fluid viscosity was set to $1e^{-4}$, and the stiffness for the pressure computation to $10$. For training data, we use columns with a particle count from $1$ to $40$ over $100$ time steps, where the boundary consists of two particles. For the evaluation, we use a subset of $10$ scenes from the data set and additionally generate $5$ scenes with $1$ to $5$ particles in free fall with a starting height of $1cm$. For the explicit solver used as a comparison in the evaluation, we use the method by Premžoe et al. [32], with the same settings as for the iterative solver, apart from a smaller time step of $0.25ms$. This was necessary as the simulations were not stable with a larger time step.

**WBC-SPH Data Set** This data set is based on the WBC-SPH solver by Adami et al. [1]. We use a particle radius of $0.005m$ and a time step of $2.5ms$. The scenes consist of randomly generated fluid volumes and obstacles with a static, square-shaped outer boundary. The fluid particles are simulated over $3200$ frames, with gravity having a random magnitude and direction for each scene. The gravitational strength can be up to $1.5g$. The randomization of gravity generates data with high variance and allows for a high degree of generalizability, e.g., fluid simulation without gravity or with other external forces than earth's gravity. With this setup, we generate $50$ scenes for training, $10$ scenes for validation, and $5$ scenes for the test data set. In our test data set, two simulations of a hydrostatic tank with a liquid height of $10cm$ and $25cm$ are added for diversification, as well as two simulations of colliding fluid drops, once with and once without gravity.

**WaterRamps Data Set** The `WaterRamps` data set is based on an MPM solver [18] and stems from GNS [37]. The data is two-dimensional, and the particle radius is twice as large at $0.01m$ compared to the `WBC-SPH` data set, with a time step of $2.5ms$.

**Liquid3d Data Set** For the 3D evaluations we use the `Liquid3d` data set from Ummenhofer et al. [48]. The data set is based on the DFSPH solver [4] with a particle radius of $0.025m$, and a time step of $0.02s$. Hence, the data is sampled more coarsely than the other two data sets, while the time step is $8$ times as large. A large time step makes the data difficult to learn because the discrepancy between the input to our network and the targeted reference becomes larger after a position update based on the integration of the external forces. Thus, the network must learn a much larger correction. In addition, the data set is three-dimensional, which introduces more degrees of freedom and additional complexity. This makes the data set a suitable and challenging environment to evaluate for our method.

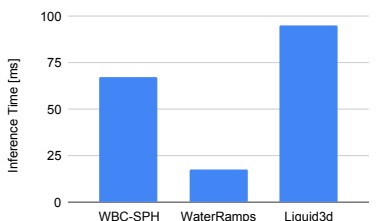

(a) Average inference time for single frames.

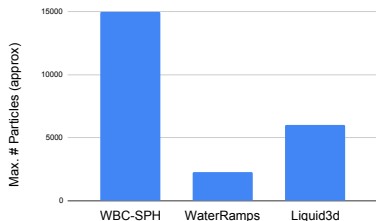

(b) Approximate maximum number of particles per randomized initialization for each data set.

Figure 15: Runtime (a) for our method and the number of particles (b) for the used datasets.

## A.4 Additional Results

**Evaluation of Performance** Figure 15a shows the average execution time of our network for the inference of a single frame of simulation for each data set. It is noticeable that the inference time for the 2D `WBC-SPH` data set is larger than for the other 2D case. This behavior is caused by the fact that there are significantly more particles in one frame of the high-resolution data set `WBC-SPH`, as shown in Figure 15b. Hence, the performance directly correlates with the number of particles that needs to be processed. In addition, we evaluated the inference performance of the different approaches. As can be seen in Figure 16, the performance correlates with the model sizes, our model having 0.47m parameters, GNS with 1.59m, and CConv with 0.18m. The smallest model, CConv, is the fastest with 2.57ms. Our model has the second fastest inference time with 10.98ms and is almost three times faster than GNS with 30.63ms. Thus, our model provides the best tradeoff between inference time and accuracy among the three methods.

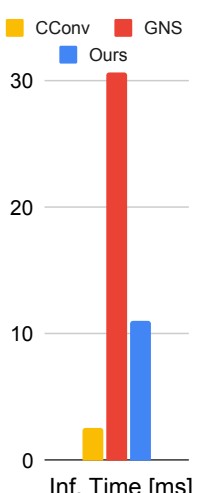

Figure 16: Runtime for different models.

**Robustness** We also evaluate our method's robustness to input noise and sampling density. First, we measure the change in EMD for varying input noise. The noise is normally distributed with a standard deviation based on the particle radius. The evaluation is based on the `WaterRamps` data set, and we also evaluate the GNS with noise as a comparison. We perturb the input particles' position with the noise and evaluate the model with these perturbed inputs. Since GNS needs six input frames, we add the noise to all input frames. We keep the noise per particle constant for all six frames so that the velocity and acceleration derived by GNS from the input frames are not affected. In line with this treatment, we do not perturb the input velocity for the ASCC model. We show the evaluation results in Fig. 17. It can be seen that both models perform equally well in the

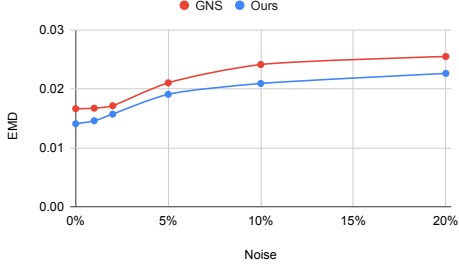

Figure 17: Inference accuracy for varying amounts of input noise. The standard deviation of the noise is expressed as a percentage of the particle radius.

Figure 18: Accuracy of a constrained (blue) and unconstrained (red) model for varying training data set sizes. The first data point corresponds to 0.39% of the training data while 100% corresponds to the complete training data set as described in Sec. A.3.

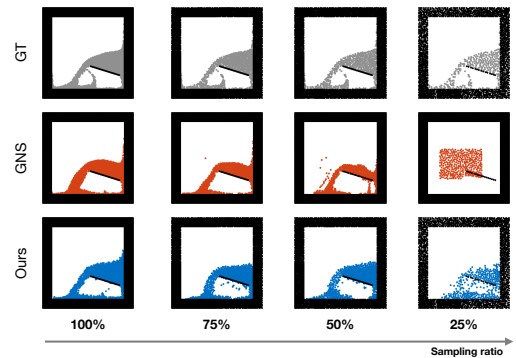
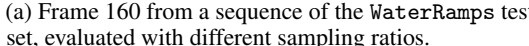

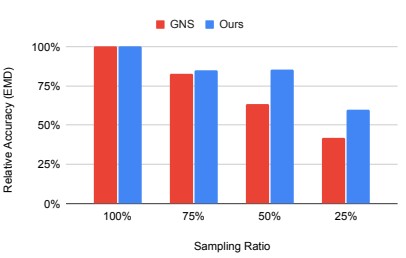

(a) Frame 160 from a sequence of the `WaterRamps` test set, evaluated with different sampling ratios.

(b) Relative accuracy of the models with different sampling ratios. The accuracy is evaluated in relation to the accuracy with a ratio of 100%.

Figure 19: Qualitative and quantitative results of the sample efficiency evaluation.

presence of noise. Even with a considerably strong relative noise of 20%, reasonable accuracy is still achieved. It is important to note that we only evaluate the first 50 frames for this comparison, as this time span is where the effects of the noise are most noticeable.

As a second test, we evaluate our model with data with a different sampling density than at training time. For this, we subsample the test data with different sampling factors, which reduces the number of input particles. We refer to this as the *sampling ratio* in the following, where a sampling ratio of 100% corresponds to the data with the original sampling density. Reducing the number of particles reduces the number of neighbors when evaluating CConv, and correspondingly down-scales the output of the convolutions. To counteract this behavior, we multiply the kernels of the CConv with the subsampling factor. For comparison, we also evaluate a GNS with inputs with different sampling ratios. Here, we multiply the accumulated value of the edge features by the subsampling factor in the GNN to compensate for the reduced amount of neighbors. The results are given in Fig. 19. As can be seen, our model maintains high accuracy of 85% up to a subsampling factor of 2. GNS, on the other hand, performs worse. While our model still produces meaningful results even with a sampling ratio of 25%, the GNS does not manage to respond correctly to particle-based boundaries. The liquid volume falls in the wrong direction and through the obstacles. It is important to note that GNS processes the square border as an implicit representation, which is not affected by the sampling. Thus, the change in sampling density does not affect the fluid-border interaction, which artificially boosts the performance of the GNS.

**Sample Efficiency**   Incorporating inductive biases typically leads to improvements in terms of sample efficiency. To evaluate this aspect, we train two variants of models, one with an anti-symmetric constraint and one without the constraint, with different subsets of the training data and compare the resulting accuracy. The results in Figure 18 show EMD as a function of the relative training data size for the *WBC-SPH* data set. It is noticeable that the constrained method performs better throughout all tests. With a training data size of ca. 6%, the constrained model achieves a performance similar to the unconstrained approach with more than 20% of the data. This highlights the advantages of our constraints for conservation of momentum in terms of sample efficiency.

**Evaluation Details** Below, we provide additional qualitative results as well as tables with numerical values corresponding to the graphs shown in the main paper.

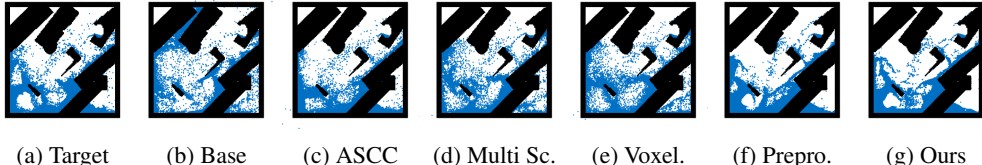

(a) Target    (b) Base    (c) ASCC    (d) Multi Sc.    (e) Voxel.    (f) Prepro.    (g) Ours

Figure 20: Frame 240 from a sample sequence for the ablation study. The gradual improvement in quality is clearly visible, with a big jump from voxelization to preprocessing. From then on, the results are much more stable.

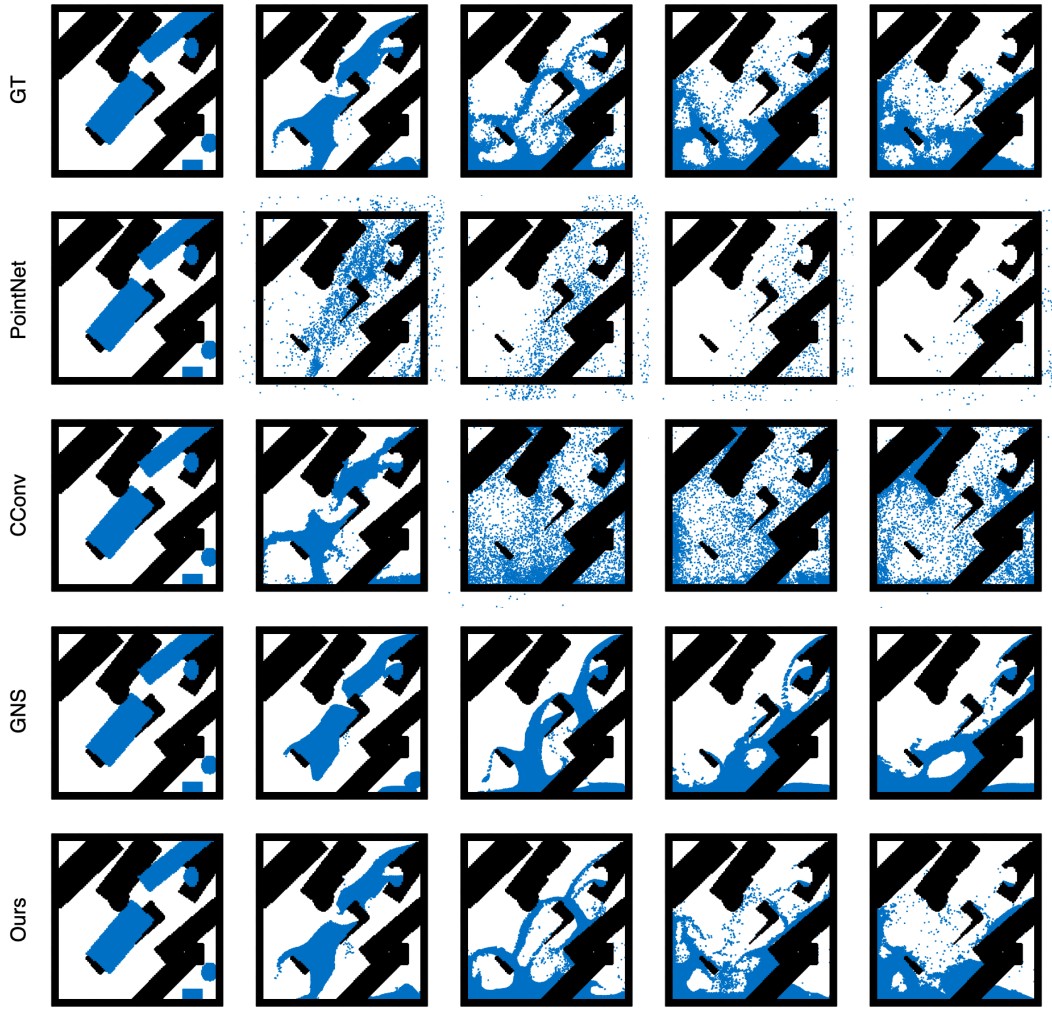

Figure 21: A test sequence from our `WBC-SPH` data set.

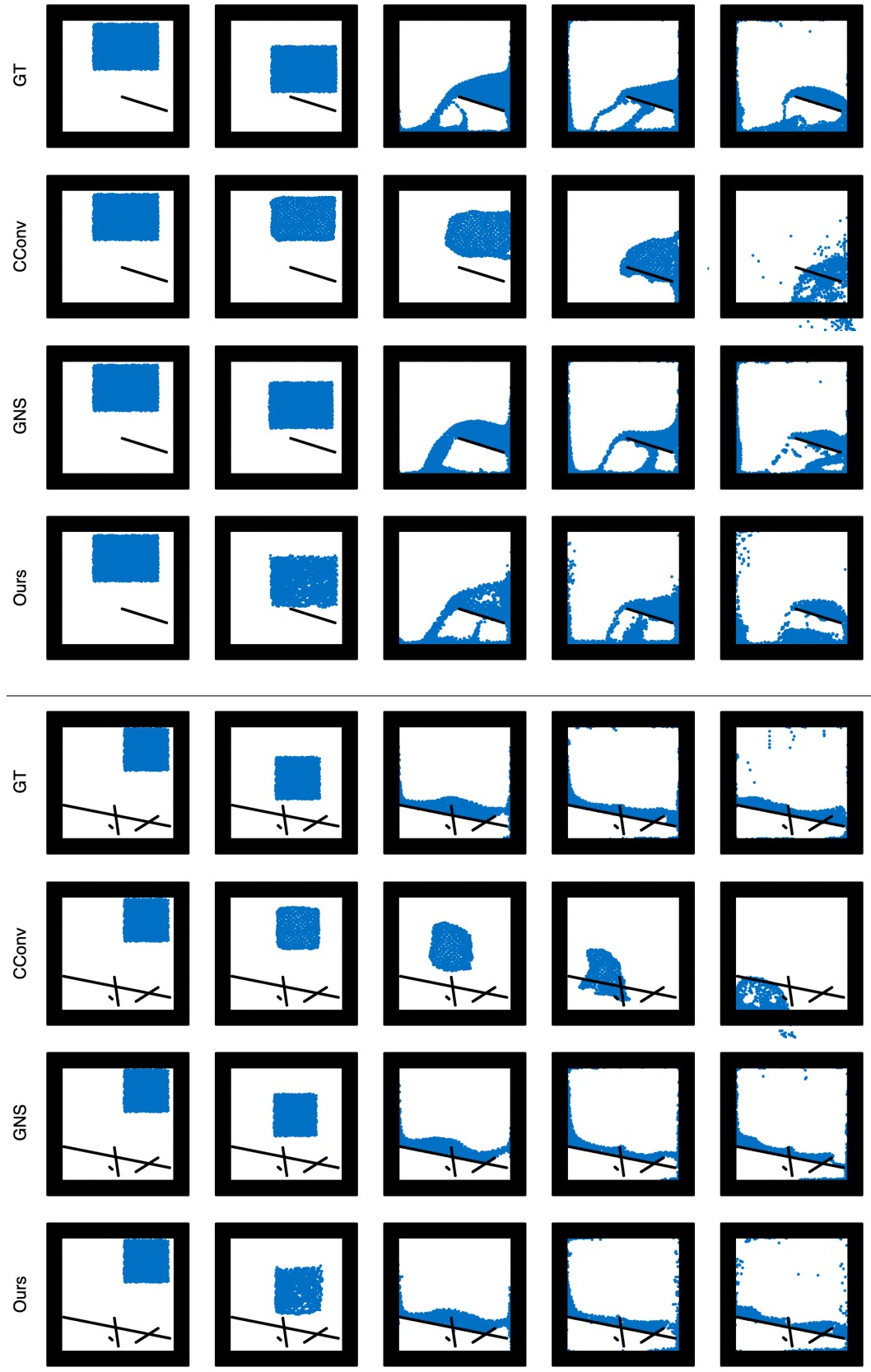

Figure 22: Test sequences with methods trained on the `WaterRamps` data set.

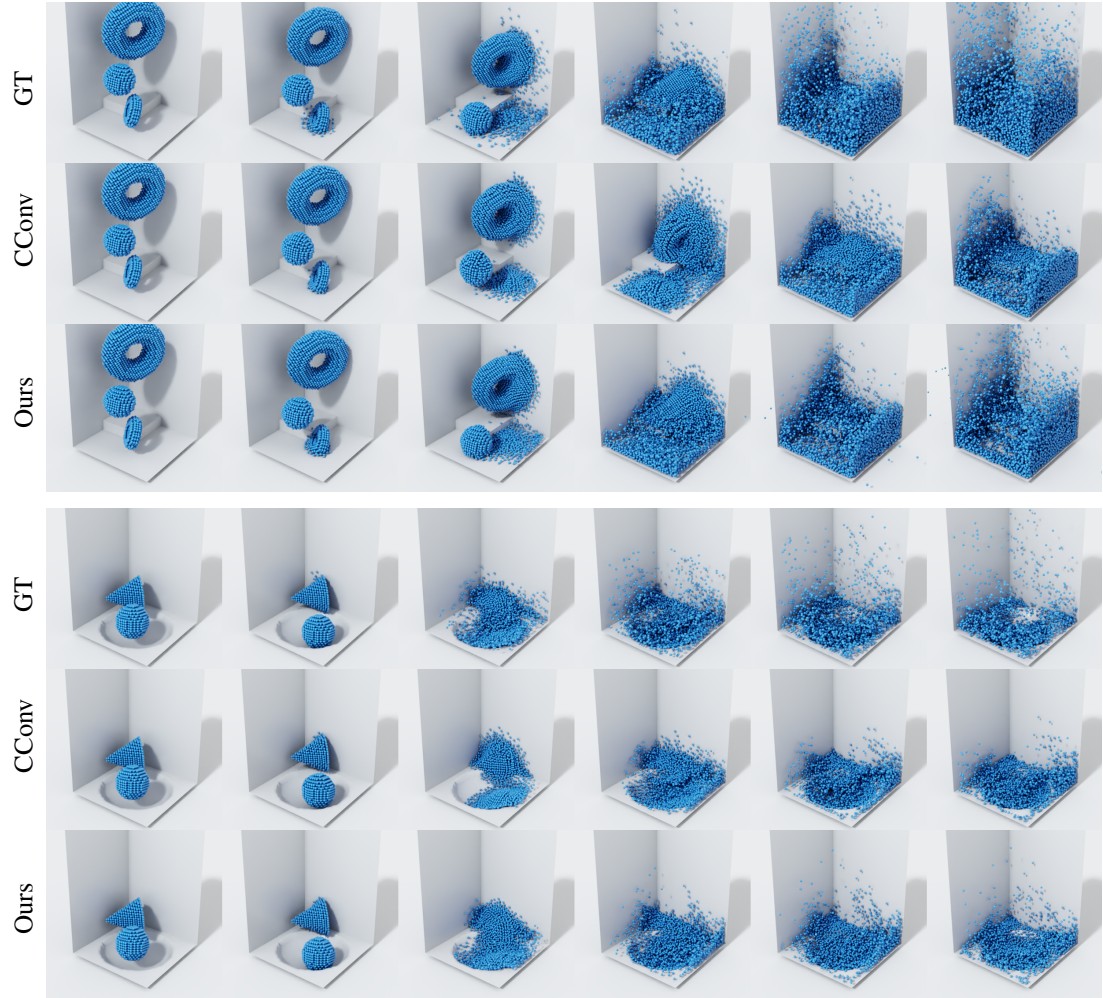

Figure 23: Test sequences based on the `Liquid3d` data set.

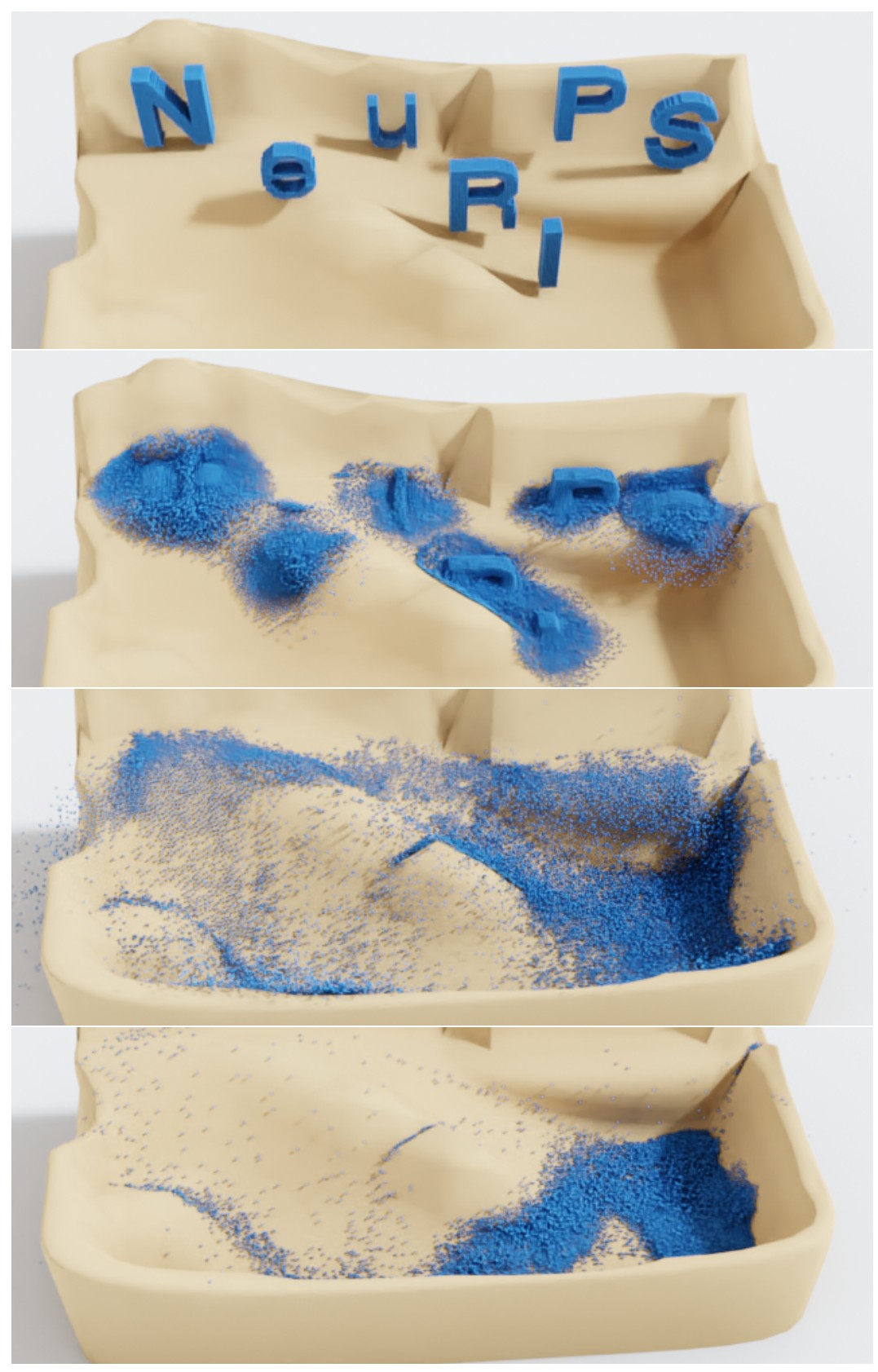

Figure 24: A complex test sequence with a $5\times$ larger particle count than the training sequences from `Liquid3d` demonstrating scalability and generalization.

|  | Column
RMSE $(\times 10^{-3})$ | Free Fall
RMSE $(\times 10^{-3})$ |
|---|---|---|
| SPH | 9.87 | 19.68 |
| No Sym. | 0.04315 | 46.36843 |
| ASCC | 0.06231 | 5.91239 |

Table 1: Quantitative evaluation based on the `Liquid Column` data set. Numbers correspond to Fig. 5 of the main paper.

|  | RMSE | Vel. Dist. | Momentum | Max. Dens. | EMD | Average |
|---|---|---|---|---|---|---|
| Base | 1.227 | 0.619 | 0.000 | 0.510 | 0.072 | 0.486 |
| ASCC | 0.964 | 0.973 | 1.000 | 0.533 | 0.185 | 0.731 |
| Multi Scale | 1.125 | 0.929 | 1.000 | 0.546 | 0.408 | 0.802 |
| Voxelize | 1.227 | 0.913 | 1.000 | 0.548 | 0.385 | 0.815 |
| Preprocess | 0.964 | 1.002 | 1.000 | 0.834 | 0.922 | 0.945 |
| Ours | 1.000 | 1.000 | 1.000 | 1.000 | 1.000 | 1.000 |

Table 2: Quantitative evaluation for the ablation study. Numbers correspond to Fig. 6 of the main paper.

|  | Random Gravity | | Tank | | Two Drops | | Overall | |
|---|---|---|---|---|---|---|---|---|
|  | RMSE | EMD | RMSE | EMD | RMSE | EMD | RMSE | EMD |
| PointNet | 0.11 | 47.3579 | 0.04 | 15.5390 | 0.07 | 1.9133 | 0.06 | 6.455210 |
| CConv | 0.16 | 221.5037 | 0.1 | 242.06389 | 0.21 | 9.4910 | 0.17 | 120.637445 |
| GNS | 0.1579 | 0.31152 | 0.512 | 0.2952 | 0.184 | 0.2628 | 0.1445 | 0.301795 |
| Ours | 0.12 | 0.134 | 0.05 | 0.00975 | 0.055 | 0.0117 | 0.07 | 0.041795 |

Table 3: Quantitative evaluation based on the `WBC-SPH` data set. Numbers correspond to Fig. 7 of the main paper. RMSE was multiplied with $10^{-3}$.

|  | RMSE $(\times 10^{-3})$ | EMD | Params. $(\times 10^{-6})$ |
|---|---|---|---|
| CConv | 0.02 | 0.29296 | 0.18 |
| GNS | 0.092 | 0.08109 | 1.59 |
| Ours 5steps | 0.11 | 0.09155 | 0.47 |
| Ours | 0.11 | 0.06156 | 0.47 |

Table 4: Quantitative evaluation based on the `WaterRamps` data set. Numbers correspond to Fig. 8, Fig. 9, and Fig. 14 of the main paper.

|  | Two Drops
EMD | Two Drops w/o Grav.
EMD |
|---|---|---|
| GNS | 0.08852 | 0.2012 |
| Ours | 0.05623 | 0.0682 |

Table 5: Quantitative evaluation of the `Two Drops` generalization test, trained with `WaterRamps` data set. Numbers correspond to Fig. 10 of the main paper.

|  | RMSE $(\times 10^{-3})$ | EMD |
|---|---|---|
| CConv | 1.69 | 0.264 |
| Ours | 2.35 | 0.2143 |

Table 6: Quantitative evaluation based on the `Liquid3d` data set. Numbers correspond to Fig. 11 of the main paper.

|  | Inference Time [ms] | Max. # Particles (approx) |
|---|---|---|
| WBC-SPH | 67.25 | 15k |
| WaterRamps | 17.62 | 2.3k |
| Liquid3d | 94.86 | 6k |
| WBC-SPH (Solver) | 10925 | - |

Table 7: Average inference time for single frames, and approximate maximum number of particles, corresponding to Fig. 15.

|       | Inference Time [ms] |
|-------|---------------------|
| CConv | 2.57                |
| GNS   | 30.63               |
| Ours  | 10.98               |

Table 8: Average inference time for single frames corresponding to Fig. 16.

| Noise Ratio | GNS (EMD) | Ours (EMD) |
|-------------|-----------|------------|
| 0%          | 0.01665   | 0.01409    |
| 1%          | 0.01673   | 0.0146     |
| 2%          | 0.01715   | 0.01573    |
| 5%          | 0.02104   | 0.01909    |
| 10%         | 0.02415   | 0.02093    |
| 20%         | 0.02551   | 0.02264    |

Table 9: Accuracy evaluation for varying amounts of input noise corresponding to Fig. 17.

| Sampling Ratio | GNS (EMD) | GNS (rel.) | Ours (EMD) | Ours (rel.) |
|----------------|-----------|------------|------------|-------------|
| 100%           | 0.09555   | 0.10309    | 100%       | 100%        |
| 75%            | 0.11544   | 0.12123    | 82.77%     | 85.04%      |
| 50%            | 0.15092   | 0.01573    | 63.31%     | 85.35%      |
| 25%            | 0.22921   | 0.12078    | 41.69%     | 59.85%      |

Table 10: Relative accuracy evaluation for different sampling densities corresponding to Fig. 19.

| Train Set Size | Unconstrained (EMD) | Constrained (EMD) |
|----------------|---------------------|-------------------|
| 0.39%          | 0.1861              | 0.08066           |
| 3.13%          | 0.0584              | 0.03881           |
| 9.38%          | 0.0424              | 0.02488           |
| 20.70%         | 0.0275              | 0.02028           |
| 100%           | 0.0280              | 0.02133           |

Table 11: Accuracy evaluation for different training set sizes corresponding to Fig. 18.