# OpenReview forum: "Guaranteed Conservation of Momentum for Learning Particle-based Fluid Dynamics"
_NeurIPS.cc/2022/Conference — NeurIPS 2022 Accept_

### Official Review · Reviewer_mZBm · 2022-07-11

**Rating:** 7
**Confidence:** 3
**Soundness:** 3 good
**Presentation:** 3 good
**Contribution:** 3 good

**Summary:**

This paper proposed the Antisymmetric Continuous Convolution that can guarantee linear momentum in learned physics simulation, and the authors demonstrated the effectiveness of this design on several particle-based simulation datasets.

**Questions:**

1. Incorporating inductive biases into neural nets can make them more sample efficient. But I believe if there are enough training data, then the unconstrained networks can also learn the desired symmetry or conservation law from the data. Do you know how much data you need to make the unconstrained model perform as well as the constrained one? A plot of how the prediction performance of both constrained and unconstrained models change across different training set sizes would be helpful.

2. In Figure 6, there is no bar for the base model on the momentum evaluation metric. Does it mean the unconstrained model does not learn the conservation of momentum at all? That seems impossible.

3. Eqn 9: shouldn't it be the integration over the neighbor of x instead of the entire P_Q?

**Limitations:**



**Strengths And Weaknesses:**

$$\textbf{Strengthes:}$$
Overall, I enjoyed reading this paper. The writing is very clear and the proposed idea is quite neat. The ablation study of the model architecture really strengthens the paper.

$$\textbf{Weaknesses:}$$
1. Lack of Baselines: I think the more SOTA models for forecasting fluid dynamics on the irregular meshes are the message-passing neural nets and the neural operator. These models could also be applied to particle-based simulations. It'd be great if the authors can include some of these models as baselines.
  * Pfaff et.al; Learning Mesh-Based Simulation with Graph Networks
  * Brandstetter et.al; Message Passing Neural PDE Solvers.
  * Guibas et.al; Adaptive Fourier Neural Operators: Efficient Token Mixers for Transformers

2. Missing related works: I think there are some other associated works about symmetry and fluid dynamics that should also be discussed in the paper such as
* Brandstetter et.al; Lie Point Symmetry Data Augmentation for Neural PDE Solvers.
* Wang et.al; Incorporating Symmetry into Deep Dynamics Models for Improved Generalization.

3. It'd be great if the authors can also evaluate the models based on computational cost or inference speed. Because I think one of the biggest advantages of using neural nets for learning complex fluid dynamics is the computation efficiency.

---

> ### Author Response · Authors · 2022-08-02
> **Response to Reviewer mZBm**
>
> Thanks for your review and the helpful feedback.
>
> **Related work**
> The works mentioned here are exciting approaches, and we agree they are relevant to our work. We have revised our paper's 'Related Work' section and discussed the proposed papers accordingly.
>
> We also agree that a detailed study with other SOTA works would be an interesting avenue for future work. We'd like to point out that the work by Pfaff et al. is very closely related to the GNS by Sanchez-Gonzalez et al., several of the authors have worked on both papers. The main difference is that the GNS is better adapted to the simulation tasks at hand, so we have focused on comparisons with this method in our submission.
>
> **Computational cost and inference speed**
> That is a good point. In the paper, we have briefly discussed the number of parameters and the training time of the individual models compared (Fig. 8), but regarding inference time, we previously only compared our method to the reference solvers (Appendix A.4).
>
> We have now performed a series of runtime comparisons between all models to assess the inference performance of the different trained models. We can see a clear correlation with the model sizes, i.e., our model with ~0.47m parameters, GNS with ~1.59m, and CConv with ~0.18m.
>
> In the evaluation, the smallest model, CConv is the fastest, with 2.57ms, while yielding the largest errors. Our model has the second fastest inference time with 10.98ms, almost three times faster than GNS with 30.63ms. Correlating these runtimes with inference accuracy, we believe that our model provides the best trade-off between performance and accuracy. We have added the evaluation and the figures for it in our revision in Appendix A.4, paragraph 'Performance'.
>
> **Impact of training set size**
> We have also performed another experiment, training constrained and unconstrained models with differently sized training sets as suggested. We found that, with a training data size of about 6%, the constrained model achieves the same performance as the unconstrained approach trained with more than 20% of the data. This illustrates the sample efficiency of our method. A more detailed description of the evaluation was added in our revision in Appendix A.4, paragraph 'Sample Efficiency'.
>
> **Figure 6 has no bar for momentum of the base model**
> We use a relative metric for the ablation study to highlight the differences between the variants. Since the momentum becomes 0 by design when the ASCC layer is added, the  relative score of the baseline model is 0 due to its momentum error of 0.095. In practice, the scores are computed with $(L_{x, final} + \epsilon) / (L_{x, variant} + \epsilon)$ with a small constant $\epsilon$ to prevent division by zero. This effectively yields 0 for the “Base” variant and 1 for all other variants. We now explain how these scores are computed in the revised version.
>
> **Equation 9**
> The neighborhood relationship is described in Eq. 9 using the kernel $g(y-x)$ with a cutoff distance that yields zero at a given distance; hence, only neighbors with non-zero weights are considered in the equation. To simplify the equation, we integrate over $P_Q$. For a practical implementation, the neighborhood is reduced to reflect the influence radius $r$ as shown in the discretized formula (Eq. 12).

---

> > ### Comment · Reviewer_mZBm · 2022-08-08
> > **Post-rebuttal Comments.**
> >
> > I want to thank the authors for the detailed answers to my and other reviewers' questions. The updated version looks much stronger now. Based on these, I'm happy to raise my score and recommend the paper for acceptance.

---

### Official Review · Reviewer_qoJP · 2022-07-12

**Rating:** 6
**Confidence:** 4
**Soundness:** 3 good
**Presentation:** 2 fair
**Contribution:** 3 good

**Summary:**

The authors present a physics-guided deep learning model for fluid simulations. The method uses continuous convolutions to process particle representations of fluids where the convolutional kernels in the final layer are constrained to be antisymmetric. The antisymmetric constraint enforces conservation of momentum. The authors show that embedding this constraint results in improved simulation results compared to other methods.

**Questions:**

- Possible typos:
  - Line 31: write out SPH
- Questions:
  - WBC-SPH: Are the strength and direction of gravity determined randomly?
  - [261] : [EMD] is especially important to reliably evaluate long simulation sequences. What is the intuition here?
  - Figure 1: In eq 2 G is a function of x_t and v_t in the figure it is a function of x’_t, v’_t.
- Comments:
  - Related work doesn't include other NN + Lagrangian methods [1][2][3][4]
  - [116] G is used to denote both the linear momentum and the network that predicts the update which is a bit confusing
  - [31] SPH is not defined until the next section
  - [219] F is used for features and forces which is a bit confusing
  - [318] “An adapted variant of PointNet” – how is it adapted?
  - [320] “For fairness…” It seems more appropriate to use the dataset of CConv since they are using a more similar methodology (CConv vs GNNs)

[1] Saemundsson, Steindor, et al. "Variational integrator networks for physically structured embeddings." International Conference on Artificial Intelligence and Statistics. PMLR, 2020.
[2] Lutter, Michael, Christian Ritter, and Jan Peters. "Deep lagrangian networks: Using physics as model prior for deep learning." arXiv preprint arXiv:1907.04490 (2019).
[3] Cranmer, Miles, et al. "Lagrangian neural networks." arXiv preprint arXiv:2003.04630 (2020).
[4] Sanchez-Gonzalez, Alvaro, et al. "Hamiltonian graph networks with ode integrators." arXiv preprint arXiv:1909.12790 (2019).


**Limitations:**

A limitation implicitly noted in the conclusion is that the learned representations are not rotationally invariant. I suspect rotationally symmetric kernels will substantially reduce expressivity. It may be possible to use something like group convolutions [1].

[1] Cohen, Taco, and Max Welling. "Group equivariant convolutional networks." International conference on machine learning. PMLR, 2016.

**Strengths And Weaknesses:**

- Originality: The proposed method is an application of continuous convolutions to particle representations of fluids. The proposed intervention is to constrain filters in the final layer to be antisymmetric. The related work is adequately situated in the context of existing literature, I encourage the authors to consider adding the literature on Lagrangian neural network to related work(see questions/comments).
- Quality: The work is technically sound and the claims are well supported. A limitation implicitly noted in the conclusion is that the learned representations are not rotationally invariant.
- Clarity: For me the paper was not easy to read. E.g., I think the paragraph at [163] interrupts the flow, perhaps it is better placed at the end of the subsection. It may not be needed since my sense is the same arguments are given in the related work section. There were also notational clashes and abreviations that were never written out (see questions/comments)
- Significance: The paper addresses the issue of learning physically plausible fluid simulations which is a challenging problem with the potential to facilitate scientific discovery.

---

> ### Author Response · Authors · 2022-08-02
> **Response to Reviewer qoJP**
>
> Thank you for the detailed and constructive feedback.
>
> **Are the strength and direction of gravity determined randomly for WBC-SPH?**
> Yes, the strength and direction of the gravity were determined randomly. Appendix A.3 contains a detailed description of the different data sets and their variations.
>
> **What is the intuition of EMD to evaluate long sequences?**
> Distance metrics such as MSE are usually evaluated point-to-point. However, over long sequences, it is common for fluid particles to mix chaotically regardless of whether the general fluid behavior resembles the reference or not, leading to big artificial per-particle errors. Thus, MSE is not suitable for reliably comparing results with long sequences. EMD, on the other hand, measures the distance between two probability distributions and is agnostic to the ordering of particles. Hence, it matches the particles such that the global distance error is minimized and counteracts the assessment errors caused by mixing. Intuitively, EMD evaluates the similarity of density-weighted volumes instead of the individual particles and thus quantifies the error of the generated shape relative to the reference, neglecting small-scale changes in particle ordering.
>
> While the Chamfer distance is also commonly used, providing a computationally cheaper alternative to EMD, we found that the inclusion of particle density in the latter is a clear benefit in terms of robust assessments for fluid simulations. In the revised paper, we have explained the reasons for using EMD in more detail.
>
> **Adaptation of PointNet as a baseline comparison**
> PointNet was initially designed for classification instead of regression tasks. The input to PointNet is usually the complete point cloud, from which a single scalar/vector is generated. In our case, we treat a PointNet as a “convolution” operator and apply it to all input particles, generating an output for each particle. Additionally, we use only the neighborhood of the particle under consideration as input. A description of the PointNet setup can be found in Appendix A.2. The goal of this setup was to create a baseline that is as simple as possible for a particle-based setting, in line with a fully-connected network in other problem settings.
>
> **Use of CConv vs. GNS data sets [L320]**
> Thank you for pointing out this sentence. We intended to highlight that we evaluate the different approaches with their respective data sets for fairness, i.e., WaterRamps for GNS. We overlooked explicitly mentioning the Liquid3d data set from the CConv paper, which we used to evaluate and compare our method to the CConv approach, e.g., in Fig. 10. We clarify this treatment in the revised version of our paper.
>
> **Comments on paper**
> Thank you for the detailed feedback regarding the paper. It was very conducive, and we have incorporated all suggested changes, and we have corrected Eq. 2 to match Fig. 1.
>
> **Group equivariant convolutional network**
> Thanks for pointing out this interesting work; we cite it now in our conclusion section.

---

> > ### Comment · Reviewer_qoJP · 2022-08-08
> > **Post-rebuttal comments**
> >
> > It seems the author’s have addressed the concerns of all reviewers and that the revised presentation is clear and convincing. With consideration of the revisions and comments from other reviewers I plain to raise my initial rating to Accept.

---

### Official Review · Reviewer_bVLD · 2022-07-13

**Rating:** 7
**Confidence:** 4
**Soundness:** 3 good
**Presentation:** 4 excellent
**Contribution:** 4 excellent

**Summary:**

The paper suggests a novel inductive physical bias to guarantee the conservation of momentum. The latter is achieved by reconsidering the resampling strategies and a reformulation of a continuous convolution operation to incorporate the physical knowledge as a hard constraint to respect the symmetry of interaction of particles by construction.

**Questions:**

1/ What is alpha in equation (3) ?

2/ Why the restriction of P_D=P_Q ? and what is its implication on the expressive power of the model ?

**Limitations:**

This work doesn't have societal concerns and the authors are aware of the current limitations of the proposed work.

**Strengths And Weaknesses:**

This work is inspired by Lagrangian mechanics where fluids could be seen as a constellation of particles in interaction. Particles are represented as nodes and their interactions are represented by edges. This construction is a good starting point to make the analogy with Geometric Deep Learning (GDL) models, including Graph Neural Networks (GNNs) and points of cloud. In GDL, we distinguish isotropic and anisotropic convolutions which are widely used and well studied. However, when it comes to design a continuous convolution operation for physical problems one should look carefully at the sampling strategy (dense / sparse connectivity at different local regions), the symmetries present in the data at different physical scales and how to design a smooth kernel that propagates correctly the physical signal while preserving its physical properties.
In this study, the authors proposes a novel design of a convolution operation capable at encoding the conservation law as a hard constraint relying on a well studied continuous operation, know as "CConv" instead of incorporating this information in the loss function as soft constraint. One of the strength of this hard constraint is to physically guide the neural network in the learning process for better generalization which a crucial topic in physics.
The proposed convolutional kernel is smooth by construction and is derived from concepts in Lagrangian discretization, mainly Smoothed Particles Hydrodynamics to integrate in hard the symmetries of the underlying physical problems. This way of designing a kernel allows to conserve the physical properties "symmetries" by design while keeping the properties of convolutional operation by design, namely permutation invariance and translation equivariance. In addition to, a careful choice of sampling scheme is taken into consideration which plays also a crucial role as the choice of convolutional operations.

The weaknesses in this paper are related to the lack of comparison with challenging datasets as done in the related works, including for instance large scale meshes with challenging obstacles and geometries to show the versatility of the proposed model. Moreover,  it could be valuable and insightful to study the robustness of the proposed convolutional operation to different discretization schemes, sampling strategies and noisy data since the goal is generalization.

In general, the paper is well written, clear and easy to follow. The claims and their structure are quite clear and the mathematical variables are well defined.
The study of the related works is well done. Notably, the authors makes a relationship between different concepts in physics, machine learning, signal processing and computer vision.
The experimental study is fair and convincing even if it could be extended.

---

> ### Author Response · Authors · 2022-08-02
> **Response to Reviewer bVLD**
>
> We thank the reviewer for the positive assessment of our work; we will address the questions and concerns in the following.
>
> **Evaluation of robustness**
> We agree that a further evaluation regarding robustness is a very good extension of our submission. We have therefore done additional experiments to evaluate the robustness of our method to input noise and sampling density.
>
> A first evaluation applies varying amounts of input noise to our model and GNS as a comparison. In this evaluation, both models reacted equally well to the input noise. Even with a considerable amount of noise of 20% relative to the particle radius, reasonable accuracy can still be achieved. The EMD increases from 0.014 to 0.022 by a factor of ~1.61 for our method. For GNS, the EMD changes from 0.017 to 0.025, with a factor of ~1.53. Thus, our approach is robust to input noise, in line with other network architectures.
>
> As a second experiment, we evaluated our model with data exhibiting a different sampling density than seen at training time. In this evaluation, our model maintains a high accuracy of ~85% for a subsampling factor of up to 2. GNS, on the other hand, performs worse with an accuracy of ~63%. We included both experiments in Appendix A.4 in the paragraph 'Robustness' with an extended description and additional figures.
>
> **Definition of alpha ($a$) in equation (3)**
> We assume this question refers to $a$, which denotes the acceleration of the considered particle. We have added a clearer definition of $a$ and other variables to reduce further ambiguities.
>
> **Restriction of $P_D=P_Q$ (from equation 10)**
> Each connection between two points from the two sets $P_D$ and $P_Q$ has to be bidirectional so that for every edge feature $f_{xy}$ there exists a counterpart $f_{yx}$. This is only possible if the two points exist in both sets. If this is true, the anti-symmetry $f_{xy} = -f_{yx}$ guarantees that the relations between the individual points cancel each other out in the sum to conserve momentum.
>
> While limiting at first sight, this requirement is easy to uphold in practice: we treat cases with $P_D \neq P_Q$ either with subset interactions or resampling. For interactions between subsets, e.g., to handle interactions between fluid particles and obstacle particles, we can solve the problem by applying the ASCC to the union set of $P_D$ and $P_Q$ and then split the sets afterward. For the multi-scale handling of our neural network architecture, we employ resampling. Here, one can first resample the data from $P_D$ to $P_Q$ and then apply the ASCC. We now explain in the revised version of our paper why the restriction is important and how it is realized in our implementation.
>
> **Evaluation with challenging data sets**
> An evaluation with additional challenging data sets is indeed a good idea and poses no inherent difficulties for our model. A central motivation for the WBC-SPH data set was to train models that can accurately handle challenging configurations with a high variance in external forces and geometric configurations. The test scenes from this data set and the large-scale scene from Fig. 21 already show that our trained model can handle highly challenging and previously unseen fluid and obstacle configurations. Since the results of the large-scale scene were easy to overlook in the previous version, we have explicitly mentioned it in the revised paper. If the reviewer can point us to specific cases we should run, we'd be happy to include additional examples in the next revision of our paper.

---

### Author Response · Authors · 2022-08-02
**General Response**

We want to thank all reviewers for their efforts to review our submission and for the numerous constructive suggestions to improve our paper. We were happy to see that the feedback was generally positive, that our method has been well received, and that our experiments have been seen as fair and convincing.

To account for the questions and queries, we have uploaded a revised PDF version with changes highlighted in green. We have extended our revision with several experiments and additions based on the requests. We consider them all valuable contributions and are very grateful for the suggestions.

Among others, the revised version includes the following changes:
- We show the robustness of our method with two new evaluations, specifically, robustness to input noise and sampling density (cf. Appendix A.4, 'Robustness'). Here, our method is fully on-par with existing work.
- In A.4 'Performance', we have included a comparison of inference runtime, demonstrating that our approach outperforms the other methods in terms of accuracy per computation.
- To show how our symmetric constraint increases sample efficiency, we evaluated the model with differently sized data sets, comparing it to an unconstrained network (in the A.4 'Sample Efficiency' section).
- Based on reviewer feedback and questions, we have revised the paper to avoid future ambiguities and incorporated the requested changes.

Below, we give more detailed answers to the individual questions of reviewers.

---

### Meta-Review · Area_Chair_iDjd · 2022-08-27

**Recommendation:** Accept
**Confidence:** Certain

**Metareview:**

The paper models fluid particle dynamics with continuous convolutions where the convolutional kernels in the final layer are constrained to be antisymmetric. This physical constraint enforces conservation of momentum.  Reviewers think this is a well-written paper. Related work suggested during the review period should be added.

**Award:**

No

---

### Decision · Program_Chairs · 2022-09-14

Accept